# Physicochemical Properties of Ti³⁺ Self-Doped TiO₂ Loaded on Recycled Fly-Ash Based Zeolites for Degradation of Methyl Orange

Iván Supelano García [1], Carlos Andrés Palacio Gómez [2,*], Marc H. Weber [3], Indry Milena Saavedra Gaona [1], Claudia Patricia Castañeda Martínez [4], José Jobanny Martínez Zambrano [4], Hugo Alfonso Rojas Sarmiento [4], Julian Andrés Munevar Cagigas [5], Marcos A. Avila [5], Carlos Rettori [5,6], Carlos Arturo Parra Vargas [1] and Julieth Alexandra Mejía Gómez [2]

1   Grupo Física de Materiales, Universidad Pedagógica y Tecnológica de Colombia (UPTC), Avenida Central del Norte 39-115, Tunja 150003, Boyacá, Colombia
2   Grupo GIFAM, Universidad Antonio Nariño, Carrera 7 #21-84, Tunja 150001, Boyacá, Colombia
3   Institute of Materials Research, Washington State University, Pullman, WA 99164-2711, USA
4   Grupo de Catálisis-UPTC, Universidad Pedagógica y Tecnológica de Colombia (UPTC), Avenida Central del Norte 39-115, Tunja 150003, Boyacá, Colombia
5   Centro de Ciências Naturais e Humanas, Universidade Federal do ABC (UFABC), Santo André 09210-580, SP, Brazil
6   Instituto de Física "Gleb Wataghin", Universidade Estadual de Campinas (UNICAMP), Campinas 13083-859, SP, Brazil
*   Correspondence: carlospalacio@uan.edu.co; Tel.: +57-608-7447569

**Abstract:** The extensive production of coal fly ash by coal combustion is an issue of concern due to its environmental impact. TiO₂-zeolite composites were synthesized, at low cost, using recycled coal fly ash from a local thermoelectric power plant to produce the zeolite using the hydrothermal method. TiO₂ was loaded by means of the impregnation method using ethanol and titanium isopropoxide between 8.7 and 49.45 wt% TiO₂. The samples were characterized by X-ray diffraction, Raman, electron spin resonance, high-resolution transmission electron microscopy, N₂ adsorption-desorption, doppler broadening of annihilation radiation, and diffuse reflectance techniques, and the photocatalytic activity of the composites was evaluated according to the degradation of methyl orange under UV light. The results show that TiO₂ crystallizes in the anatase phase with a Ti³⁺ oxidation state, without post-treatment. TiO₂ particles were located within the pores of the substrate and on its surface, increasing the surface area of the composites in comparison with that of the substrates. Samples with TiO₂ at 8.7 and 25 wt% immobilized on hydroxysodalite show the highest degradation of methyl orange among all studied materials, including the commercial TiO₂ Degussa P25 under UV light.

**Keywords:** aluminosilicate; coal fly ash; dye degradation; photocatalyst; TiO₂; zeolite

## 1. Introduction

Pollution by organic compounds originating from industrial waste waters results in a major threat to human health and aquatic life. To solve this issue, the metal oxide-based semiconductor photocatalysts are considered as a solution due to their capacity to promote oxidation–reduction processes under electromagnetic radiation [1,2]. Among the photocatalysts, the TiO₂ is one of the most extensively studied materials due to its physicochemical properties, low cost, abundance, low toxicity, strong oxidation ability, ease of preparation, and stability [3]. To enhance the photocatalytic properties of TiO₂ via light irradiation, procedures, including doping with metal and nonmetal ions and the insertion of intrinsic defects, such as Ti³⁺, to introduce states within the TiO₂ band gap, have been developed

to produce modified $TiO_2$ nanoparticles [3]. However, $TiO_2$ exhibits some undesired characteristics, such as a low adsorption capacity, a high aggregation of nanoparticles, and difficult reclamation, which reduce its photocatalytic activity [4]. Particle suspension and immobilized $TiO_2$ systems have been designed to overcome the above limitations [2,5]. The immobilization of $TiO_2$ particles on a larger particulate substrate is expected to be the most efficient photocatalysis system from an economical and performance viewpoint [5]. Substrates favoring the adsorption of pollutants, accelerating the photocatalytic rate, can avoid the agglomeration of $TiO_2$ particles, facilitate its recovery from an aqueous medium, and can significantly reduce the recombination of electron-hole pairs in $TiO_2$ [4]. Substrates with a porous structure, such as clays, activated carbon, zeolites, and silica, have been used, resulting in better photocatalytic performance [2]. Efficient dye degradation by $TiO_2$ loaded on aluminosilicates depends on the adsorption processes, the intraparticle diffusion of dye molecules, light penetration, higher surface areas, adsorbed hydroxyl groups, dye and catalyst surface potential, dye concentration, the crystalline structure, the hydrophobicity of the substrate, and the type of Si/Al ratio, in the case of zeolites, among other factors [4–8]. In regards to the zeolites, commercially available [6,9–13] and natural [4,14,15] types are the most common employed for use as substrates for the loading of $TiO_2$ particles.

On the other hand, the large-scale production of those systems increases the cost due to the fact that they may require the use of specific templates or surfactants using high temperature treatments [5]. A low cost material used as substrate to immobilize $TiO_2$ particles is the zeolite synthesized from coal fly ash (CFA). CFA is a fine powder produced during coal combustion as a by-product; its chemical composition comprises amorphous Si and Al, with crystalline phases of quartz, mullite, and iron oxides, among others [16–18]. CFA is considered a pollutant, and to reduce its environmental impact, CFA has been used as a Portland cement replacement in cement-based products, as a precursor for geopolymers, and as a raw material for aluminosilicate synthesis [17,19–24]. Aluminosilicates, such as zeolite from CFA, can be obtained by alkaline activation using hydrothermal treatment [17,25–27]. Synthesis parameters of the hydrothermal treatment, including temperature, activation period, and CFA properties, including chemical composition and $SiO_2/Al_2O_3$ ratio, are factors affecting the resulting products [17].

According to the revised literature, most of the research uses commercial and natural zeolites to load $TiO_2$ particles. However, fewer papers reported the CFA-based zeolite to support $TiO_2$ [5,28].

In previous works, coal fly ash from a local thermoelectric power plant was activated with NaOH by the hydrothermal method, and the effects of the concentration of NaOH, time of activation, and temperature of the synthesis on the formation of zeolite were evaluated [17,29]. In this work, two of those samples were selected to immobilize $Ti^{3+}$ self-doped $TiO_2$ on zeolite by the impregnation method, in which the amount of $TiO_2$ was varied between 8.7 and 49.45 wt%. The photocatalytic activity of the composites was evaluated using methyl orange (MO) in an aqueous medium under UV radiation.

## 2. Results

### 2.1. Properties of CFA

Figure 1a depicts the SEM micrograph of CFA; it shows the morphology in which the major particles are spherical in shape, corresponding to cenospheres. Minor particles form agglomerates that correspond to carbonaceous particles. Figure 1b shows the X-ray diffraction pattern of CFA, which was indexed with quartz, mullite, maghemite, and hematite as major crystalline phases, with PDF cards: 01-078-2315, 01-083-1881, 00-039-1346, and 00-024-0072, respectively. The broad XRD signal at $2\theta < 35°$ indicates the presence of amorphous content. The chemical composition of CFA, determined by X-ray fluorescence, shows that the CFA contains mainly $SiO_2$ (62.57 (0.45)%), $Al_2O_3$ (24.62 (0.50)%), $Fe_2O_3$ (5.77 (0.82)%), $K_2O$ (1.48 (0.04)%), $TiO_2$ (1.37 (0.05)%), CaO (1.27 (0.07)%), MgO (0.53 (0.04)%), and $Na_2O$ (0.46 (0.51)%), with an $SiO_2/Al_2O_3$ ratio of 2.5.

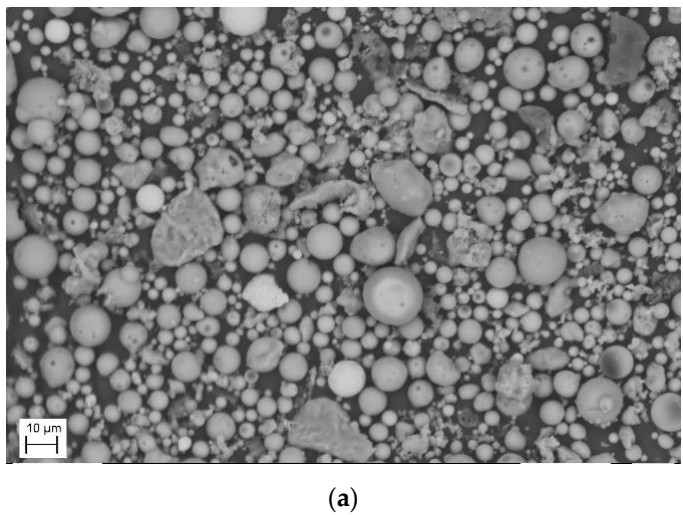

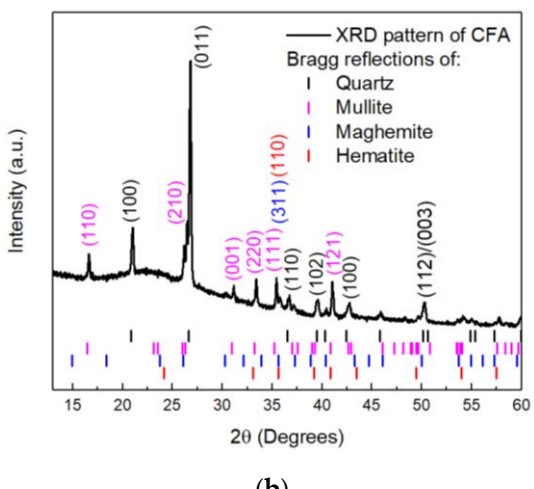

(**a**)                    (**b**)

**Figure 1.** (**a**) SEM image of CFA, (**b**) XRD pattern for CFA indexed with quartz, mullite, maghemite, and hematite.

### 2.2. Structural Properties of Synthesized TiO$_2$ and Substrates

Figure 2 shows the XRD patterns for TiO$_2$, and the H and C substrates. All diffraction peaks for TiO$_2$ were indexed with a I 41/amd space group of anatase (ICSD 24276). The XRD data of the substrates was analyzed by Rietveld refinement, and the identified crystalline phases are listed in Table 1.

**Table 1.** Crystalline phases identified in H and C substrates.

| Substrate | Phase | Chemical Formula | Space Group | ICSD Card No. | Percentage of Phase (%) |
|---|---|---|---|---|---|
| H | Hydroxysodalite | $Na_8(AlSiO_4)_6(OH)_2$ | P -4 3 n | 60840 | 59 (1.30) |
| | Zeolite P1 | $H_{24}Al_6Na_6O_{44}Si_{10}$ | I -4 | 9550 | 21 (1.87) |
| | Mullite | $Al_{2.272}O_{4.864}Si_{0.728}$ | Pbam | 100805 | 18 (0.80) |
| | Aluminosilicate | $H_{3.92}Al_{1.92}Na_2O_{12}Si_{3.08}$ | I -42d | 067210 | 2 (0.12) |
| C | Cancrinite | $Na_8(AlSiO_4)_6(CO_3)(H_2O)_2$ | P 63 | 20334 | 68 (0.92) |
| | Aluminosilicate | $NaAlSiO_4$ | P 32 | 433181 | 32 (0.72) |

Hydroxysodalite and zeolite P1 correspond to zeolite materials and are the major phases identified in H substrate. Cancrinite is a zeolite material and is the major phase identified in C substrate. According to the DRX results, it can be concluded that the alkaline activation of CFA by NaOH and posterior heat treatment allows for the obtaining of zeolite materials.

### 2.3. Properties of TiO$_2$-Zeolite Composites

Figure 3a,b shows the XRD patterns for the HT and CT composites. In both HT and CT composites, the Bragg reflections of the anatase TiO$_2$ phase and the Bragg reflections of the major phases of the substrates can be distinguished.

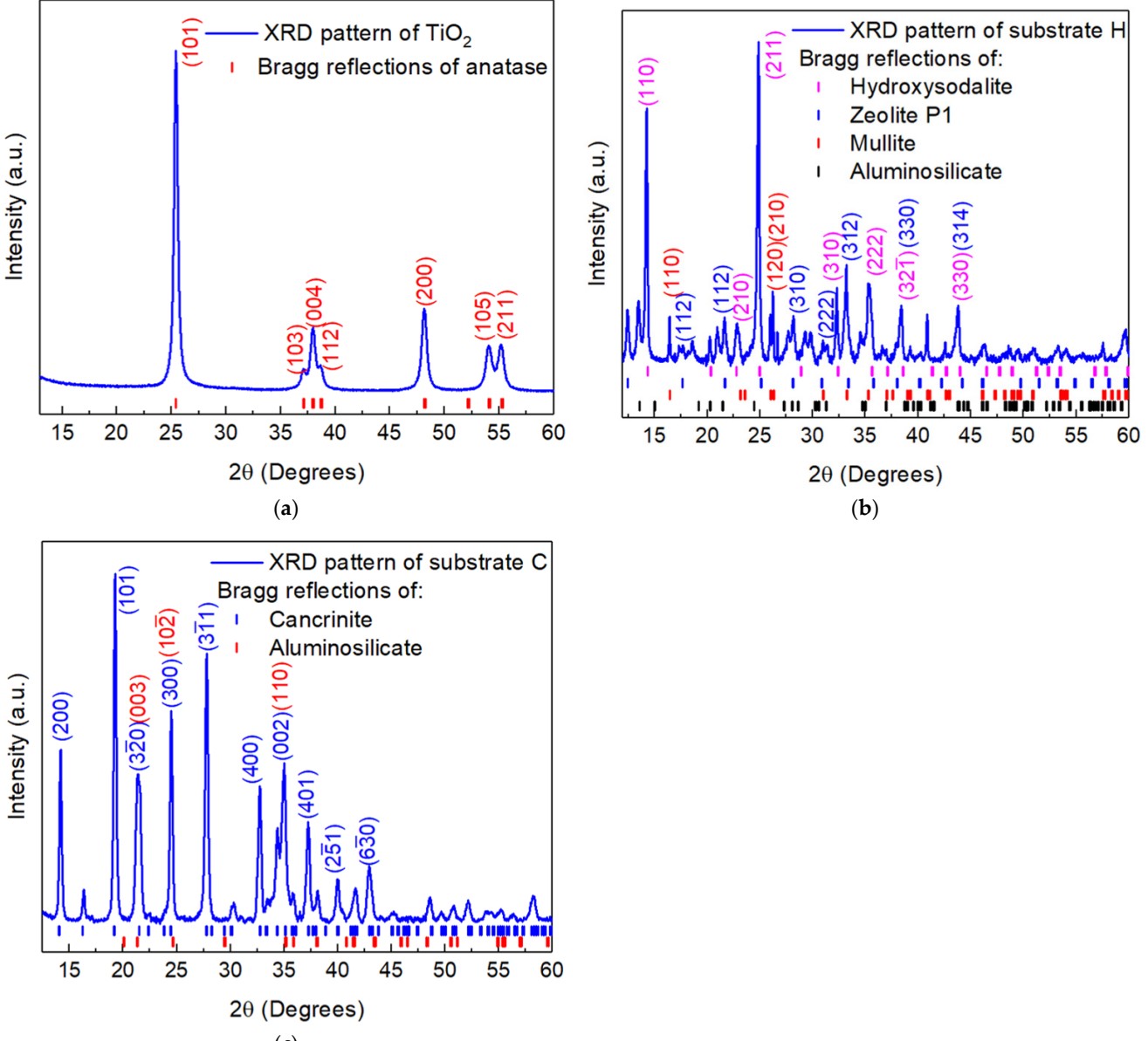

**Figure 2.** XRD pattern for (**a**) synthesized TiO$_2$ indexed with anatase phase, (**b**) H substrate indexed with hydroxysodalite, zeolite P1, mullite, and sodium aluminosilicate, (**c**) C substrate indexed with cancrinite and an aluminosilicate.

For HT composites, the intensity ratio I$_{(101)TiO2}$/I$_{(211)H}$ of the major reflection of anatase (101), located at around of 25.4°, and hydroxysodalite (211), located at around of 24.8°, was calculated; correspondingly, for CT composites, the intensity ratio I$_{(101)TiO2}$/I$_{(101)C}$ of the major reflection of anatase (101), located at around of 25.4°, and cancrinite (101), located at around of 19°, was calculated (Figure 4a). Those ratios tend to increase when the amount of TiO$_2$ increases in both the CT and HT composites. These results suggest that the incorporation of TiO$_2$ as anatase was successful on both the H and C substrates. For TiO$_2$ anatase, in both CT and HT systems, the *a* and *c* lattice parameters were calculated from the (200) and (101) reflections, respectively, using the equations:

$$\lambda = 2d_{hkl}\sin(\theta) \tag{1}$$

$$1/d^2_{hkl} = (h^2 + k^2)/a^2 + l^2/c^2 \tag{2}$$

where λ is the wavelength of the incident X-rays, $d_{hkl}$ is the interplanar distance for a tetragonal system, h, k and l are the Miller indices, and *a* and *c* are the lattice parameters.

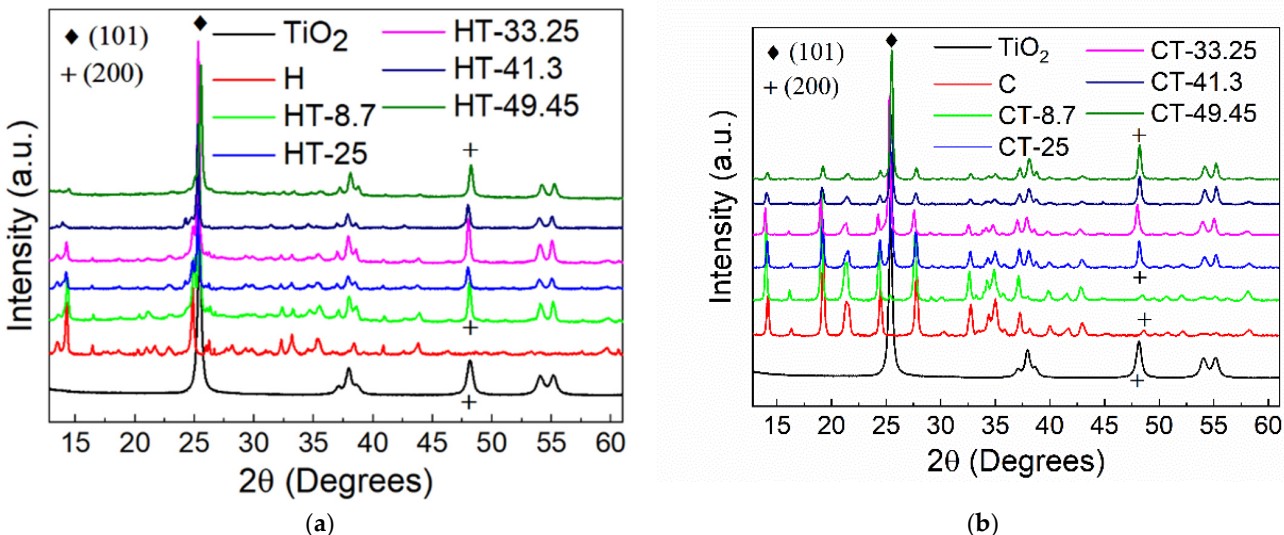

**Figure 3.** XRD pattern for (**a**) HT composites (**b**) CT composites. For comparative purposes, the XRD of C, H, and synthesized TiO₂ are depicted.

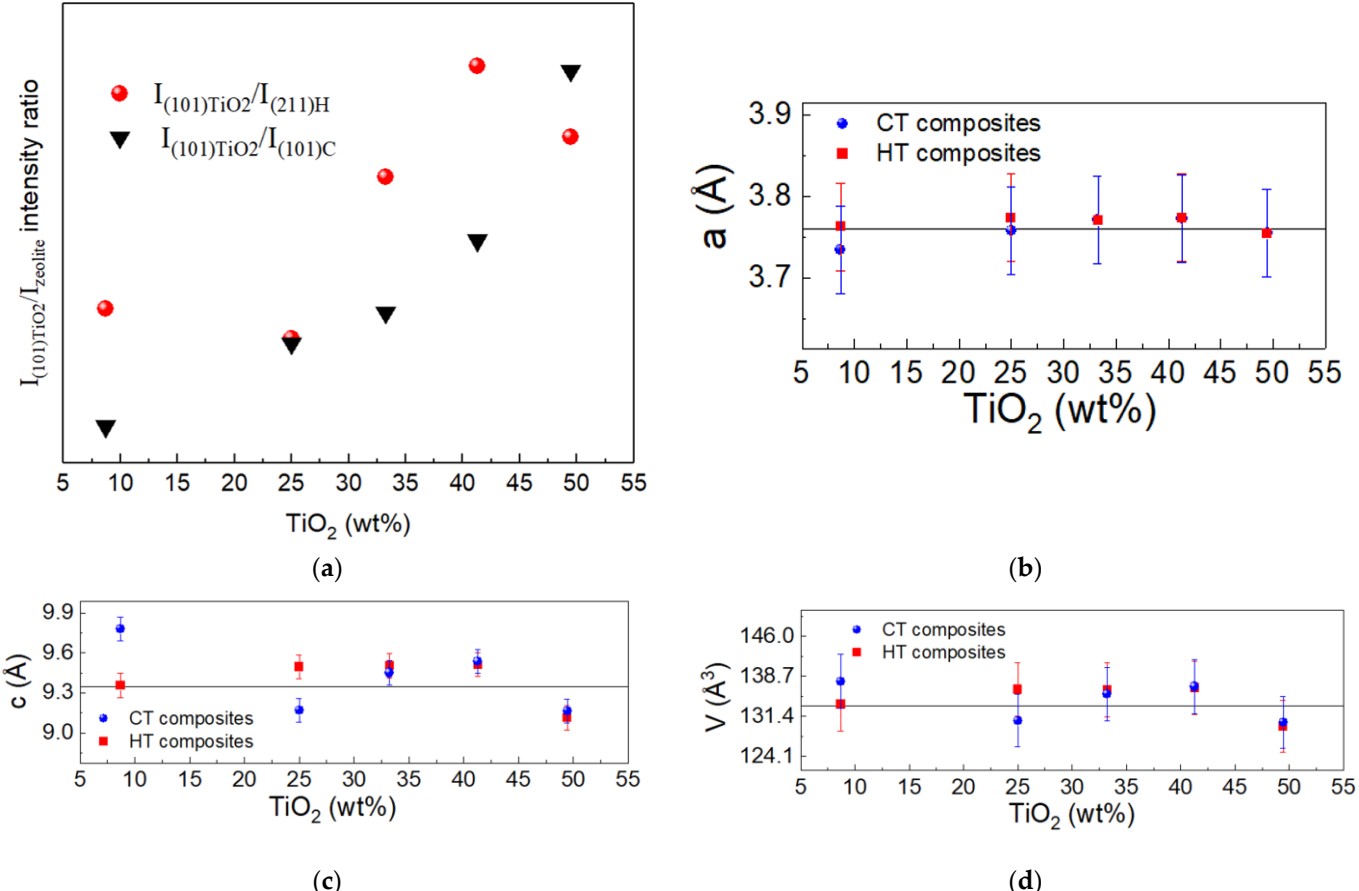

**Figure 4.** (**a**) $I_{(101)TiO2}/I_{(211)H}$ and $I_{(101)TiO2}/I_{(101)C}$ intensity ratios. (**b**) Lattice parameters a and (**c**) c; and (**d**) cell volume. The continuous line represents the value for synthesized TiO₂.

Figure 4b–d depicts the lattice parameters and cell volume of $TiO_2$ for the HT and CT composites. The results show that lattice parameter *a* remains constant in both the CT and HT composites. The lattice parameter *c* and cell volume V exhibit a slight variation for 8.7, 25, and 49.45 wt% $TiO_2$ in comparison with the other samples. The crystallite size for immobilized $TiO_2$ in all the samples was calculated using the Williamson–Hall method, which considers the contribution of crystallite size ($\beta_D$) and microstrain ($\beta_S$) to the broadening of a peak, ($\beta_{hkl}$) as [30]:

$$\beta_{hkl} \cos\theta = 4\varepsilon\sin\theta + K\lambda/D \tag{3}$$

where $\beta_{hkl}$ is the full width at half maximum, K is a constant, $\lambda$ is the wavelength of the X-ray, $\varepsilon$ is the microstrain, and D is the crystallite size; the latter can be calculated from the analysis of the plot of $\beta_{hkl}\cos\theta$ vs. $4\sin\theta$. The crystallite size values are listed in Table 2; these values are larger in the HT composites than in the CT composites, indicating that the substrate H favors the growth of $TiO_2$.

**Table 2.** Morphological and optical parameters for HT and CT composites.

| Sample | $TiO_2$ Crystallite Size (nm) * | Eg (eV) ** | G *** | ΔH (G) *** | Relative Number of Spins ($\times10^{15}$) *** | SBET ($m^2/g$) **** | Desorption Average Pore Diameter (nm) **** | Total Pore Volume ($cm^3/g$) ($\times10^{-2}$) **** |
|---|---|---|---|---|---|---|---|---|
| HT-8.7 | 73.8 | 3.31 (2) | 1.999 (4) | 3.2 | 0.22 | 15 | 15 | 4 |
| HT-25 | 64.5 | 3.31 (4) | 2.000 (4) | 3.8 | 1.33 | 10 | 12 | 2 |
| HT-33.25 | 76.6 | 3.84 (2) | 1.998 (7) | 3.9 | 1.06 | 10 | 14 | 3 |
| HT-41.3 | 88.6 | 3.84 (2) | 1.999 (8) | 8.7 | 0.40 | 16 | 8 | 3 |
| HT-49.45 | 42.3 | 2.85 (1) | 1.998 (5) | 7.8 | 1.55 | 13 | 8 | 3 |
| H | – | – | – | – | – | 11 | 24 | 4 |
| $TiO_2$ | 46.8 | 2.99 (2) | 2.000 (4) | 8.3 | 2.43 | 28 | 4 | 3 |
| CT-8.7 | 48.1 | 3.50 (3) | – | – | – | 6 | 24 | 2 |
| CT-25 | 53.5 | 3.64 (2) | 1.999 (3) | 8.3 | 0.27 | 8 | 13 | 2 |
| CT-33.25 | 74.1 | 3.92 (7) | 2.000 (6) | 8.3 | 0.26 | 8 | 12 | 2 |
| CT-41.3 | 31.5 | 3.44 (7) | 1.999 (4) | 8.3 | 0.06 | 9 | 15 | 2 |
| CT-49.45 | 33.6 | 3.68 (3) | 1.999 (4) | 7.8 | 0.04 | 7 | 12 | 2 |
| C | – | – | – | – | – | 6 | 25 | 2 |

* Calculated from XRD patterns using the Williamson–Hall method. ** Calculated from UV-Vis DRS spectra using the modified Kubelka–Munk equation. *** Calculated from ESR data. **** Calculated from $N_2$ adsorption-desorption isotherms.

The structure of the anatase in all samples was verified by Raman measurements. Figure 5a,b shows the Raman spectra for the HT and CT composites, respectively; for comparison, the Raman spectra of synthesized $TiO_2$ is shown. All samples depict the characteristic five peaks of the $D_{4h}^{19}$ *I 41/amd* space group of the anatase around 144, 197, 399, 519, and 638 $cm^{-1}$, corresponding to Eg(1), Eg(2), B1g(1), an overlap of B1g(2) with the A1g and Eg(3) optical modes, respectively [31]. The majority of the samples exhibit a long-range crystalline order, except for CT-8.7. For both the HT and CT composites, a smooth variation of the Raman shift and the linewidth of the most intense Raman peak around 143 $cm^{-1}$, in comparison with the values of synthesized $TiO_2$, are depicted (Figure 5c,d). These values are similar in both CT and HT for each of the values of the immobilized $TiO_2$, except for 25 wt%. According to the literature, the change in the peak position and linewidth are related to defects in the stoichiometry, the quantum size effects, the presence of secondary phases, and the influence of the substrate [32–34]. In this work, the results suggest that the change in the peak position and linewidth are mainly due to $TiO_2$ quantity.

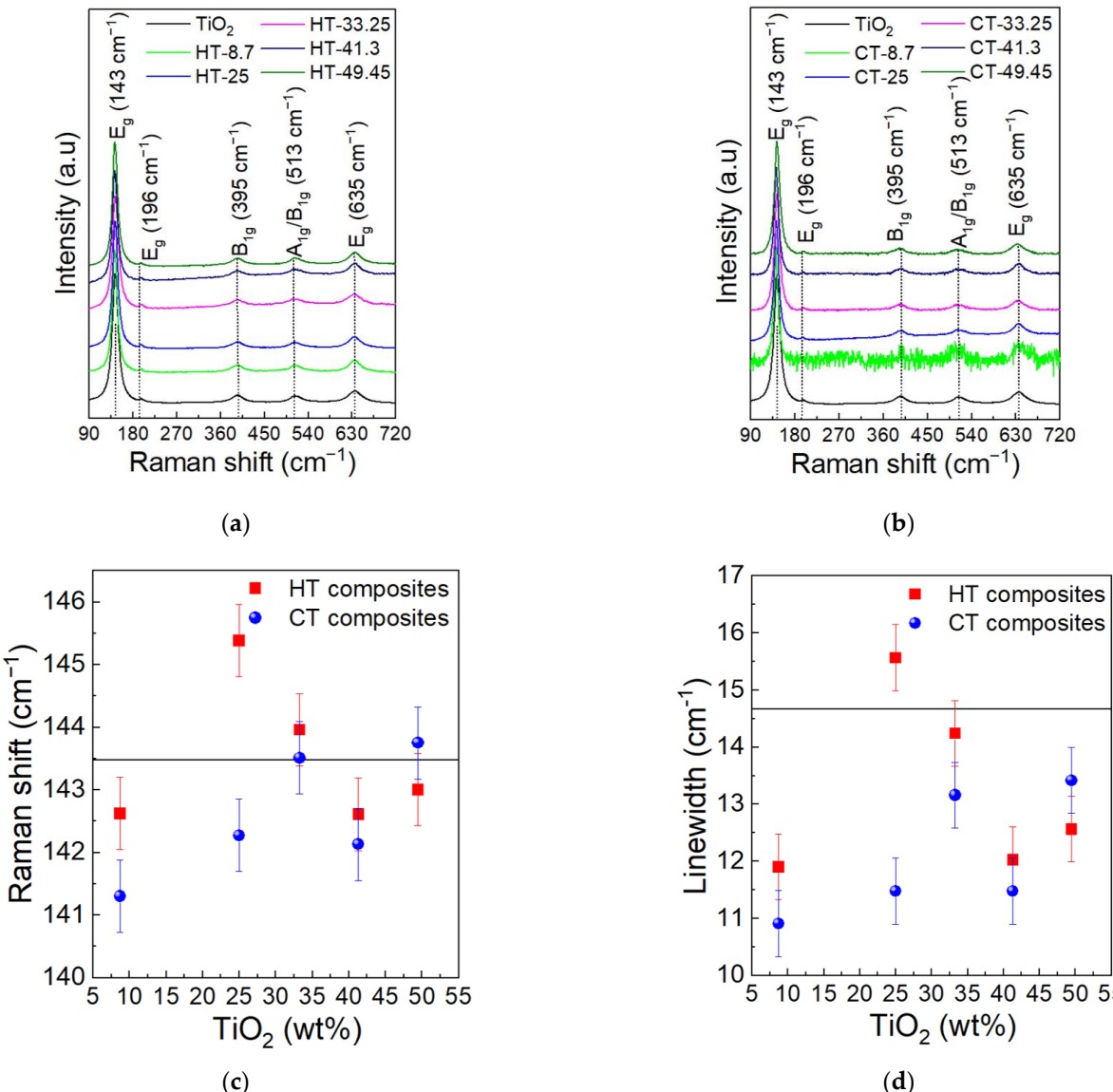

**Figure 5.** Raman shift spectra for (**a**) HT composites, (**b**) CT composites, (**c**) variation of Raman shift, (**d**) linewidth for the 144 cm$^{-1}$ peak as a function of TiO$_2$ weight percentage. the continuous line represents the parameters for synthesized TiO$_2$.

Figure 6a,b show the ESR spectra at room temperature between 500 G and 550 G for the substrates, composites, and synthesized TiO$_2$. The spectra for H and C substrates show a characteristic resonance for a g value of 2.124 and 2.135, respectively; these are related to the presence of Cu$^{2+}$ in the substrate matrices [35]. For synthesized TiO$_2$, a resonance at g = 2.000 is observed; this is related to the Ti$^{3+}$ electronic configuration due to oxygen vacancies [36–40]. For both the HT and CT composites, the contributions to the ESR spectra from the substrate and TiO$_2$ related to the Ti$^{3+}$ state are observed (Figure 6c,d). Table 2 lists the g value of the resonance, ΔH that corresponds to the distance between peaks (line width), and the relative number of spins calculated by integrating the ESR signal $dI/dH$ twice and comparing this result with a standard sample of 0.11% of KCl with $366 \times 10^{13}$ spin/cm.

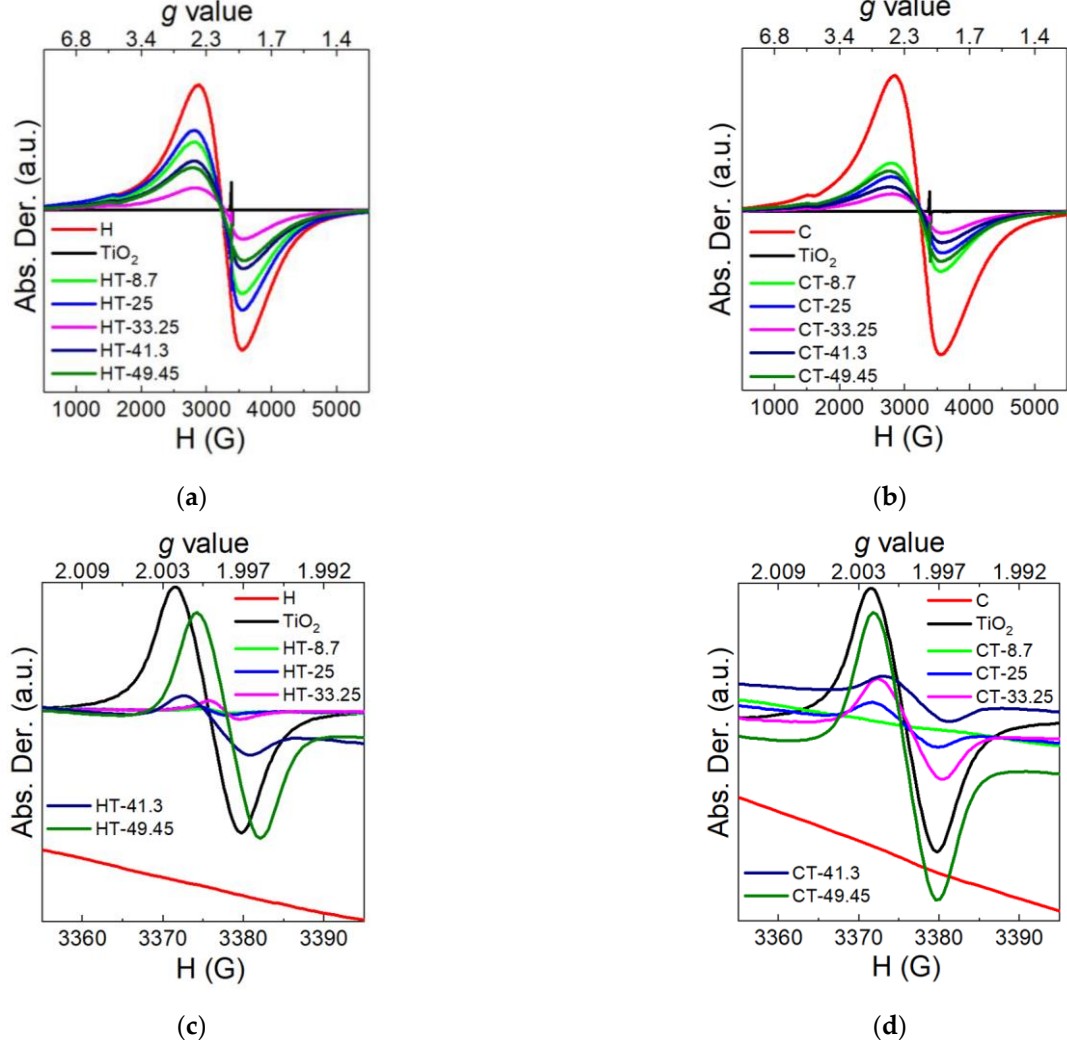

**Figure 6.** ESR spectra between 500 and 5500 G for (**a**) HT composites; (**b**) CT composites. ESR spectra around 3375 G for (**c**) HT composites; (**d**) CT composites.

According to the relative number of spins, the $Ti^{3+}$ ESR signal is higher in HT composites than in CT composites; thus, the H substrate favor the formation of $Ti^{3+}$ states in the $TiO_2$ rather than C substrate. The ESR signal tends to increase when the $TiO_2$ amount grows in both the HT and CT composites. The synthesis of $TiO_2$ with oxygen vacancies is one of the most important research topics in the chemistry of metal oxides; therefore, a wide variety of methods have been developed to produce self-modified $TiO_2$ materials [41]. Thus, it is worth noting that the samples produced in this work under the method described above allow the synthesis of $TiO_2$ with the formation of oxygen vacancies, without post-treatment. Narrow $\Delta H$ values—around 5 G reported for $TiO_2$ at low temperatures—are mainly attributed to $Ti^{3+}$ ions in the bulk, rather than on the surface of the $TiO_2$ particles [42]. In this work, the HT-8.7, HT-25, and HT-33.25 samples show a smaller $\Delta H$ value, which may indicate a larger contribution of the $Ti^{3+}$ state in the bulk than in the other samples. The presence of $Ti^{3+}$ suggests that it contributes to the observed variation in the Raman shift and the Raman linewidth, as a stoichiometry defect.

Figure 7 shows selected TEM images of the HT and CT composites with 8.7, 25, and 41.3 wt% of $TiO_2$. The results show that the $TiO_2$ particles are distributed in the zeolite surface, with sizes ranging from 3.85 nm, as in HT-25, Figure 7b, to sizes larger than 50 nm, as in HT-8.7, Figure 7a, possibly explained by agglomeration processes. The lattice spacing of around 0.256 nm assigned to (101) is found in HT-41.3, Figure 7c, and CT-41.3, Figure 7f,

samples. In CT composites, the $TiO_2$ particles are found in a wide range of sizes, ranging from 14 nm, as in CT-8.7, Figure 7d, to sizes larger than 33 nm. Lattice spacings of around 0.335 nm, 0.254 nm, and 0.208 nm, assigned to (101), (103), and (200), corresponding to anatase, were found, as shown in Figure 7d–f. From these images, it is observed that the interface between the $TiO_2$ particles and the zeolite substrates varies gradually. It is apparent that the $TiO_2$ particles were successfully incorporated onto the substrate, forming the $TiO_2$-zeolite composites. The nonhomogeneous particle size of $TiO_2$, especially those with a particle size smaller than 20 nm, makes an additional contribution to the observed variation in the Raman shift and Raman linewidth, as a quantum effect.

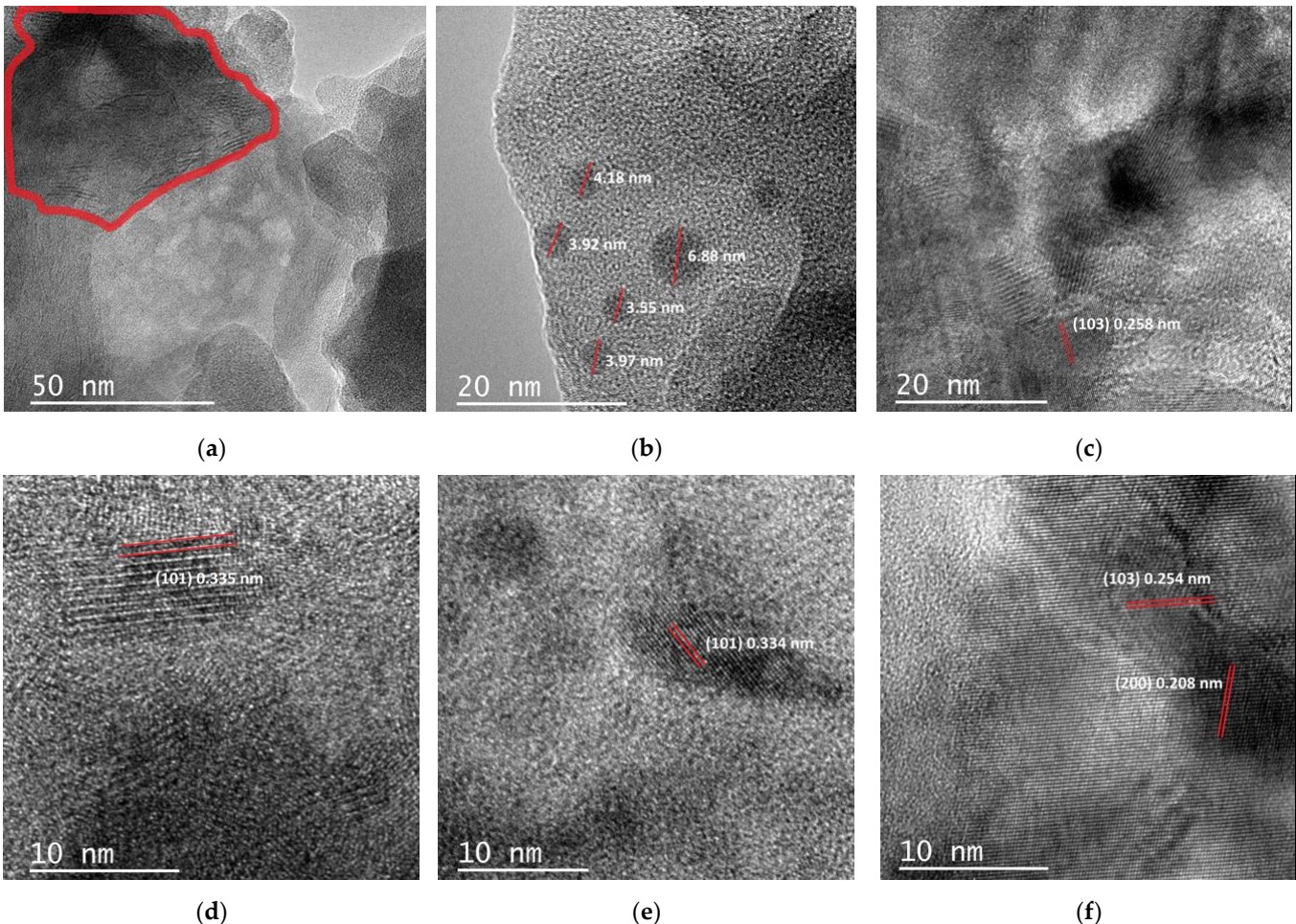

**Figure 7.** HRTEM images for (**a**) HT-8.7, (**b**) HT-25, (**c**) HT-41.3, (**d**) CT-8.7, (**e**) CT-25, and (**f**) CT-41.3. Red line represents the interface between zeolite and TiO2 in Figure 7a and the lattice spacings for Figure 7b–f.

The $N_2$ adsorption-desorption isotherms for synthesized $TiO_2$ and the substrates, HT composites, and CT composites are shown in Figure 8a–c, respectively. The curves for all samples exhibit a similar shape, which is characteristic of type IV, according to the IUPAC classification, associated with mesoporous materials. The hysteresis loops are similar for C, H, and CT/HT composites characteristics of type H3, associated with lamellar pore structures with slit- and wedge-shaped pores. For synthesized $TiO_2$, the hysteresis loop is characteristic of type H2, associated with cylindrical and spherical pores. The surface area, $S_{BET}$, was calculated using a BET equation from isotherms curves, the average pore diameter was calculated by the BJH method using the desorption branch, and the pore volume was calculated at a single point of saturation. Table 2 summarizes the results. The surface area of all CT composites is higher than that of the C substrate. The surface area in

the HT composites is higher than that of H substrate except for HT-25 and HT-33.25. In both composite systems the larger surface area was obtained for a loading of 41.3 wt% $TiO_2$. The average pore diameter decreases in both the CT and HT composites in comparison with that of the C and H substrates, respectively. The total pore volume decreases in HT composites in comparison with that of the H substrate, whereas the total pore volume increases in the CT composites in comparison with that of C substrate, except for in the CT-49.45 composite. There is a reduction in the average pore diameter in both the CT and HT composites as compared to that of the C and T substrates, respectively, probably due to pore blocking caused by $TiO_2$.

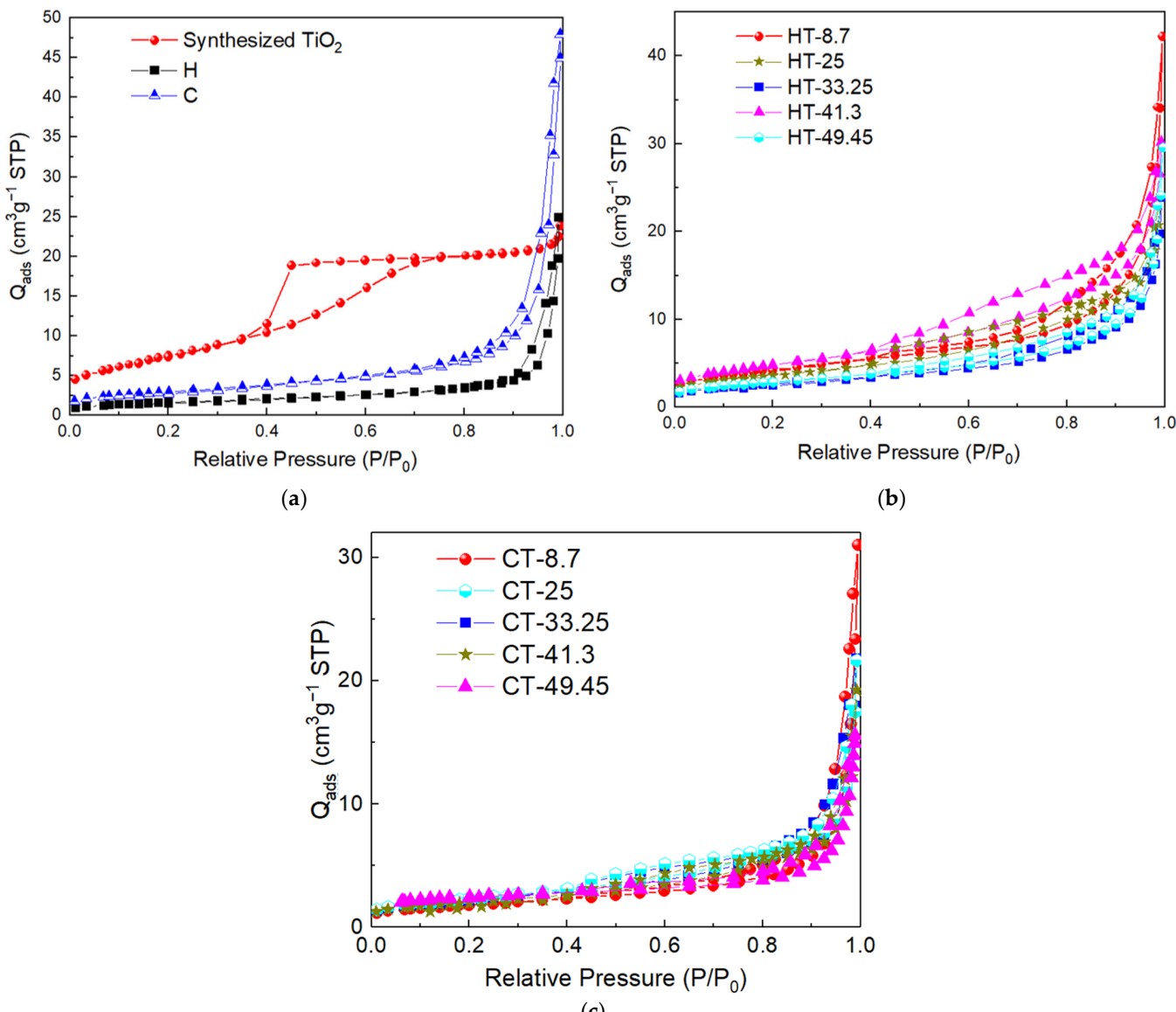

**Figure 8.** $N_2$ adsorption isotherm of (**a**) synthesized $TiO_2$ substrates; (**b**) HT composites; (**c**) CT composites.

Figure 9a,b shows the positron annihilation line shape S as a function of incident positron energy in the HT and CT composites, respectively; for comparison purposes, the response in the synthesized $TiO_2$ and the substrates is shown. The vertical line separates the surface and bulk region; the annihilation of the positrons in the bulk and/or the annihilation of the p-Ps (para-positronium) in open volumes or defects take place between 1 and 5 keV. The average of the S parameter between 1 and 5 keV is listed in Table 3. The S parameter is higher in the substrates than in the composites and synthesized $TiO_2$, and

is higher in the H than in the C substrate. In these samples, the S parameter is related to the interaction of positrons with pores. According to synthesized $TiO_2$, the value of S can be attributed to the oxygen vacancies detected by ESR spectroscopy. For the HT composites, the S parameters in synthesized $TiO_2$ is lower than in other HT composites; for these samples, the S value lies between that of zeolite and synthesized $TiO_2$. This result suggests that any quantity of $TiO_2$ deposits into the pores of the substrate and does not fill its cavities. Regarding the CT composites, the value of S for the composites is close to that of synthesized $TiO_2$. These results suggest that a larger number of pores are filled by $TiO_2$ in the H substrate in comparison with the C substrate, and a larger amount of $TiO_2$ is immobilized on the surface rather than in the pores in the C substrate in comparison with that in the H substrate. This explains the behavior observed concerning the total pore volume in the HT and CT composites.

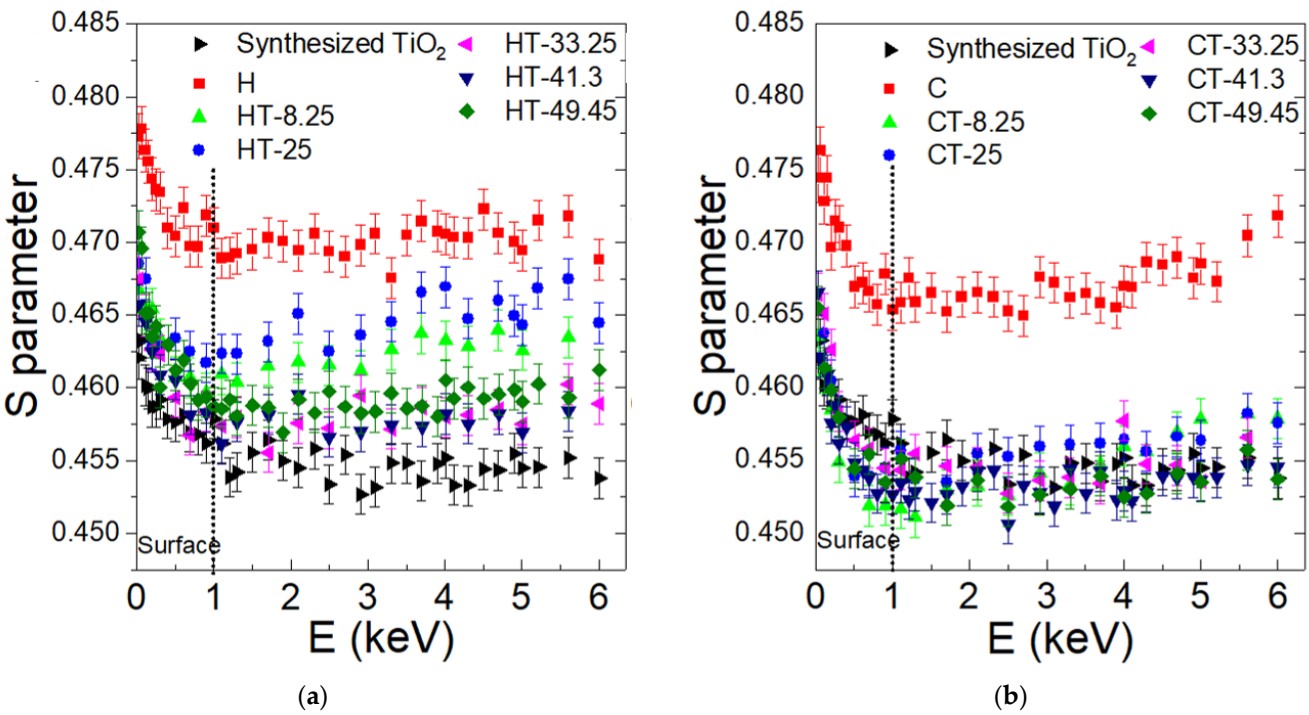

**Figure 9.** Variation with positron beam energy for the S parameter for (**a**) HT composites; (**b**) CT composites. Dashed line separates the surface and bulk region.

**Table 3.** Representative slope of the S-W plot and <S> parameter between 1–5 keV.

| Sample | Slope | <S> | Sample | Slope | <S> |
|--------|-------|-----|--------|-------|-----|
| HT-8.7 | −1.32 (24) | 0.46219 (140) | CT-8.7 | −1.81 (33) | 0.45419 (137) |
| HT-25 | −1.79 (30) | 0.46434 (137) | CT-25 | −1.62 (18) | 0.45558 (137) |
| HT-33.25 | −1.76 (18) | 0.45775 (138) | CT-33.25 | −1.95 (23) | 0.45445 (136) |
| HT-41.3 | −1.68 (14) | 0.45756 (137) | CT-41.3 | −1.90 (12) | 0.45321 (136) |
| HT-49.45 | −2.27 (14) | 0.45917 (137) | CT-49.45 | −1.86 (16) | 0.45324 (136) |
| H | −1.88 (14) | 0.46975 (138) | C | −2.24 (13) | 0.46679 (139) |

Figures 10 and 11 show the S-W plot for synthesized $TiO_2$, substrates and both composite systems, respectively. The slope of the S-W plots is useful to observe the change in the nature of the positron trapping defect. Theoretically, S depends linearly on W for each kind of defect. The lowest W value indicates the response from the surface; the higher value of W indicates the response of the bulk. For the HT and CT composites, it can be considered that all points are represented for the same slope that will indicate one kind of defect (mainly pores), since there is no variation in the positron annihilation fraction

in the trapping sites in each of the samples; as a function of incident positron energy, the dispersion is attributed to a variation of the pore sizes, which is not homogeneous. Slope values are summarized in Table 3. It is noted that all samples exhibit a small change in the slope, and there is a trend to decreases the slope when $TiO_2$ increases in the HT composites, whereas there is not a clear trend in the CT composites; this indicates that a variation exists in the positron annihilation fraction as a function of the loading of $TiO_2$ particles, which depends on the substrate; thus, it shows the changing size of the trapping sites, and thus, a change in the pores of the samples.

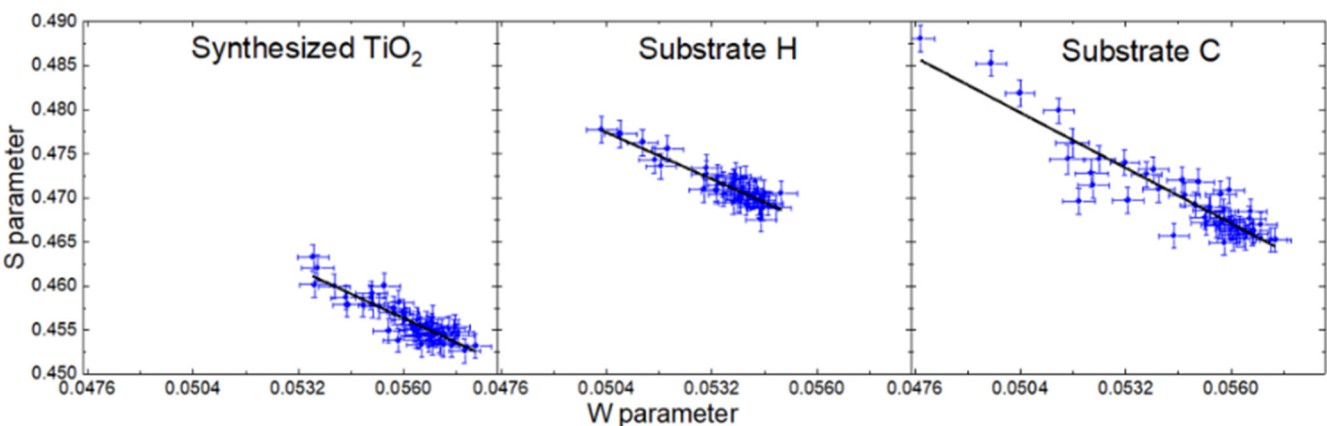

**Figure 10.** S-W plot for synthesized $TiO_2$, H, and C substrates. Continuous line represents the average slope.

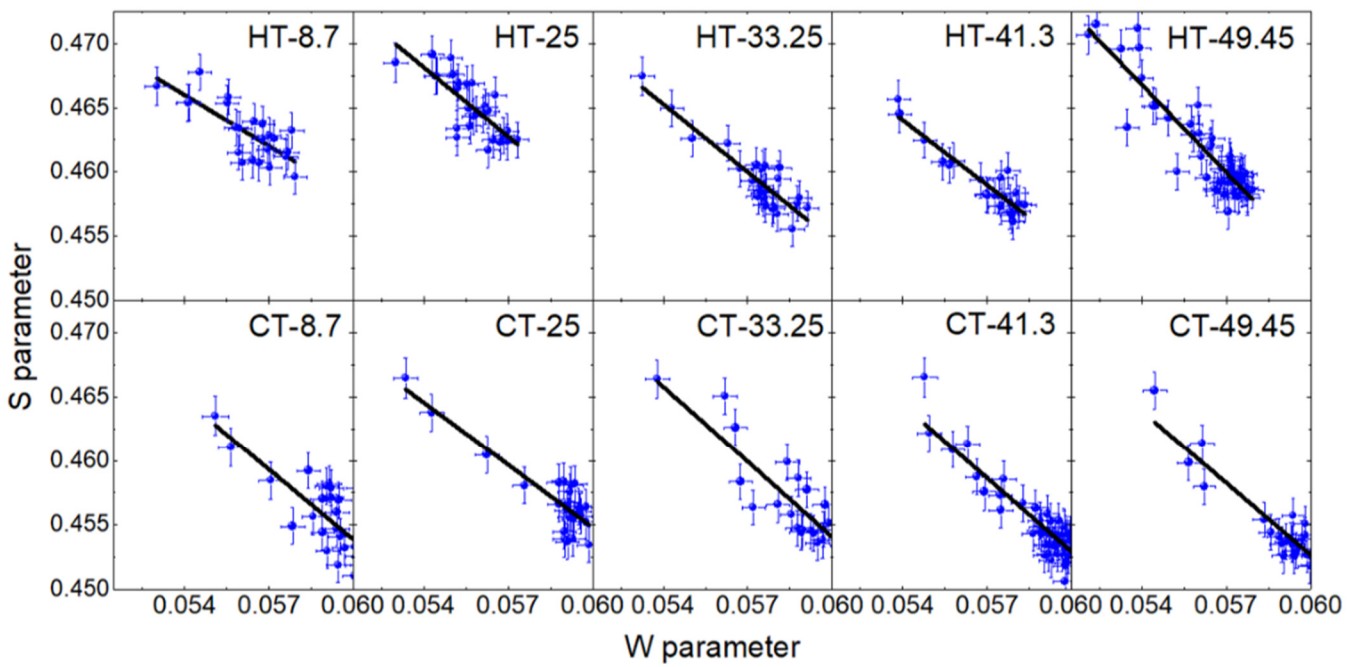

**Figure 11.** S-W plot for HT and CT composites. Continuous line represents the average slope.

Figure 12 displays the UV-Vis DRS spectra for the HT and CT composites; for comparison, the spectra for synthesized $TiO_2$ and the substrates are shown. For both HT and CT composites, the range of the reflection edge increases in comparison with that of synthesized $TiO_2$. The results suggest that the reflection edge observed in the composites is due to $TiO_2$ and is strongly affected by the interaction between $TiO_2$ and the substrate. The band gap energy ($E_g$) for the synthesized $TiO_2$ and the composites was calculated by a linear fit of the modified Kubelka–Munk equation as a function of the energy (($F(R)*h\nu)^{1/2}$ vs. $h\nu$)

and the allowed indirect transition, and the $E_g$ values are listed in Table 2. As can be seen from the table, the calculated $E_g$ for synthesized $TiO_2$ is 2.99 eV, which is lower than 3.2 eV for bulk $TiO_2$ anatase, which may be caused by the $Ti^{3+}$ electronic configuration and the crystallite size.

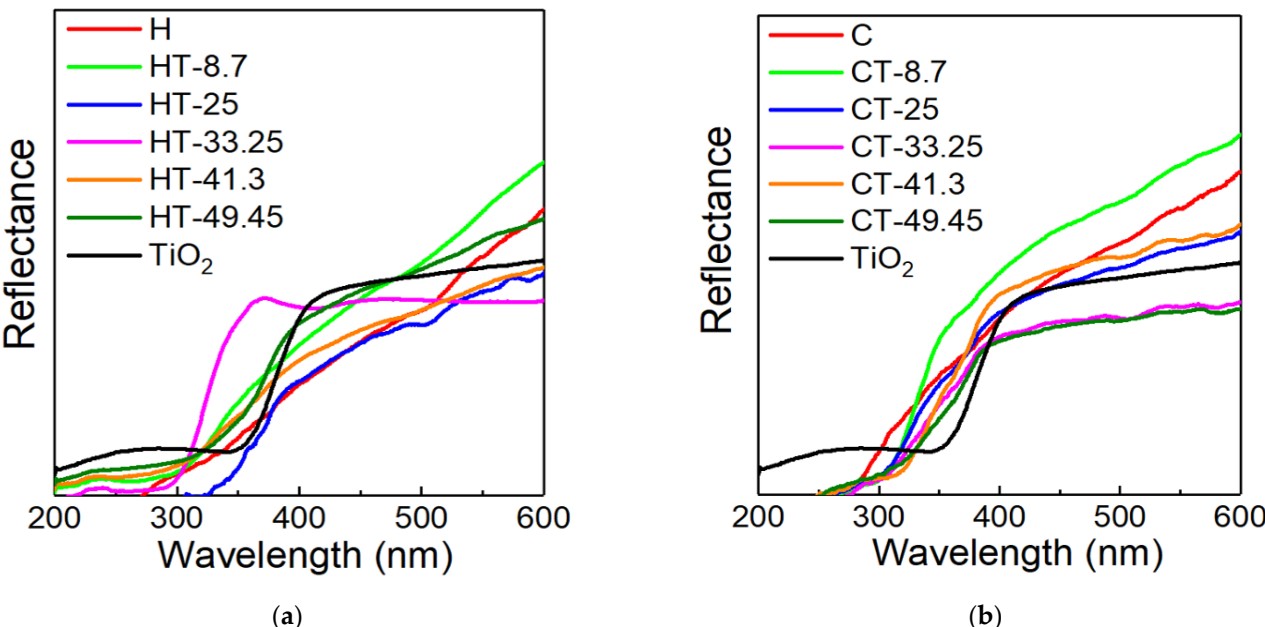

**Figure 12.** UV-Vis diffuse reflectance spectra for (**a**) HT composites; (**b**) CT composites.

Figure 13 shows UV-Vis absorption spectra of the degradation of MO using the samples HT-8.7, HT-25, synthesized $TiO_2$, and commercial Degussa P25 $TiO_2$. The spectra display two absorption bands at 270 nm and 463 nm related to the transition of the aromatic rings and the azo bonds, respectively [43], whose intensity decreases with irradiation time, indicating the degradation of MO.

Figure 14a,b shows the variation in the relative concentration ($C/C_0$) of MO as a function of the reaction time for the HT and CT composites, respectively. The composites labeled as HT-8.7 and HT-25 presented better photocatalytic behavior than other composites and the commercial Degussa P25 $TiO_2$. Thus, after 210 min of photocatalytic reaction, the HT-8.7 and HT-25 composites favored a dye degradation of 84% and 87%, respectively, compared to the 93% and 67% degradation evidenced using synthesized $TiO_2$ and Degussa P25 $TiO_2$, respectively.

On the other hand, nearly 17% of the dye was degraded using the HT-16.85 and HT-33.25 composites; 19% and 30% using HT-41.3 and HT-49.45, respectively; 3% using CT-8.7 and CT-25; 20%, 6%, 8%, and 0% using CT-16.85, CT-41.3, CT-49.45, and CT-6-425-33.25, respectively. Desorption processes shown in any samples, such as HT-33.25 and CT-33.25, can be attributed to a low generation of $^{\cdot}$OH and $O_2^{\cdot-}$ radicals and/or a high electron-hole recombination rate [44].

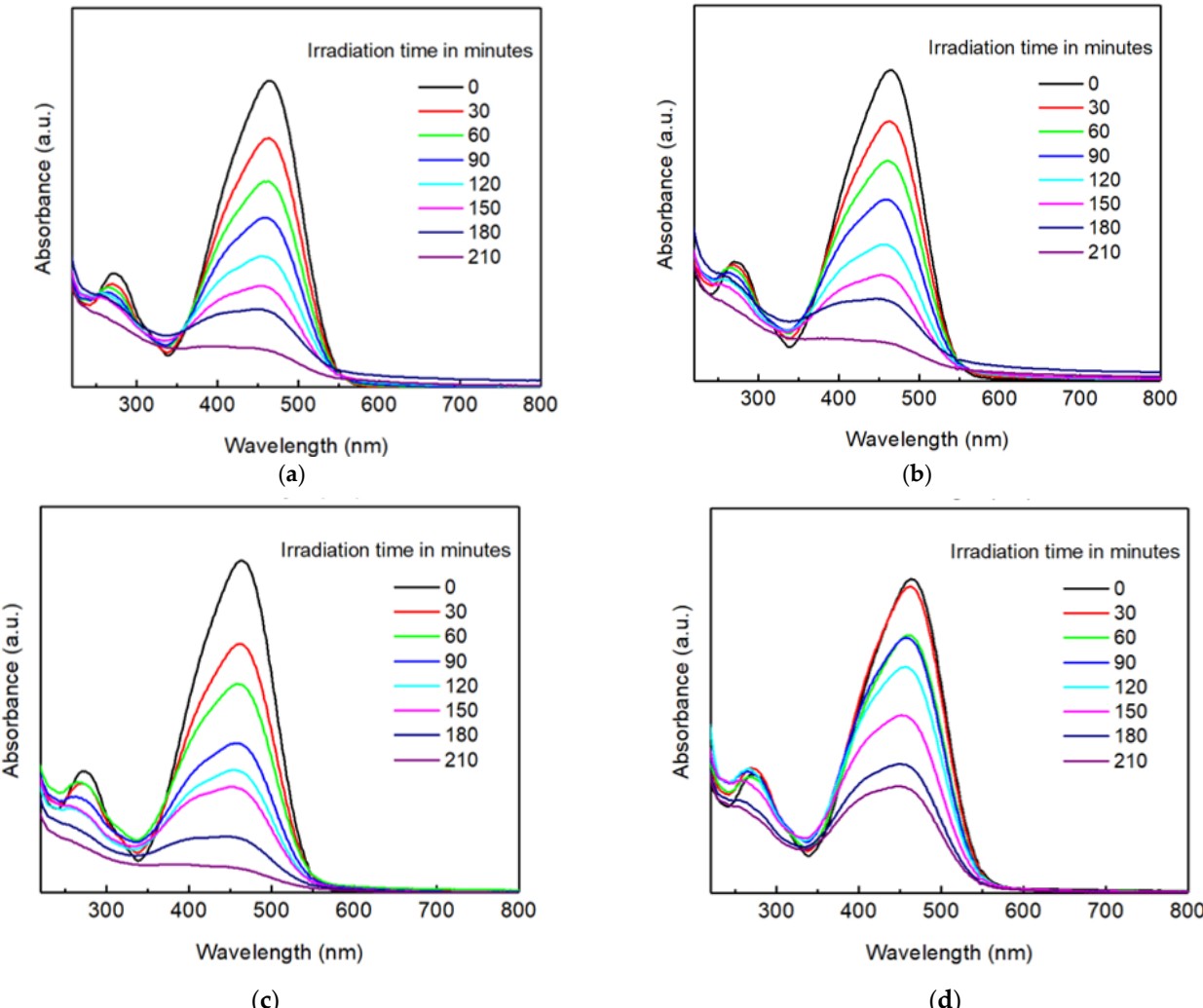

**Figure 13.** UV-Vis spectra of MO solution photoreduction by (**a**) HT-8.7, (**b**) HT-25, (**c**) synthesized, and (**d**) commercial TiO$_2$.

The total organic carbon analysis was carried out for the reactions involving the materials that evidenced the best photocatalytic behavior. The contents of TOC after 210 min of reaction were 46, 36, 15, and 64% for HT-8.7, HT-25, Degussa P25 TiO$_2$, and synthesized TiO$_2$, respectively. These results indicate that the mineralization of MO using these materials occurs in the following order: synthesized TiO$_2$ > HT-8.7 > HT-25 > commercial TiO$_2$. The higher degradation of MO using synthesized TiO$_2$, followed by the HT composites, can be explained in terms of the presence of Ti states that promote the charge separation process, acting as a holes trap [45]. However, the larger surface area and larger crystallinity of TiO$_2$ aids in the separation of the photogenerated electron and holes [4]. The degradation efficiency exhibited by the HT-8.7 and HT-25 composites can be explained in terms of a larger generation of $^{\cdot}$OH and $O_2^{-}$ radicals due to a larger irradiated surface area and a low electron-hole recombination rate [44]. Usually, the photodegradation of organic dyes by semiconducting oxides is described by the pseudo first order equation [43,46]:

$$\ln(C_0/C) = kt \tag{4}$$

where $C_0$ is the concentration of the dye at adsorption equilibrium, $C$ is the concentration of the dye at time t, and k is the apparent rate constant. The $\ln(C_0/C)$ vs. t plot is displayed in Figure 14c for the HT-8.7, HT-25, and synthesized TiO$_2$ samples, in which the values of the rate constant k are $7.60 \times 10^{-3}$, $7.50 \times 10^{-3}$, and $7.93 \times 10^{-3}$, respectively.

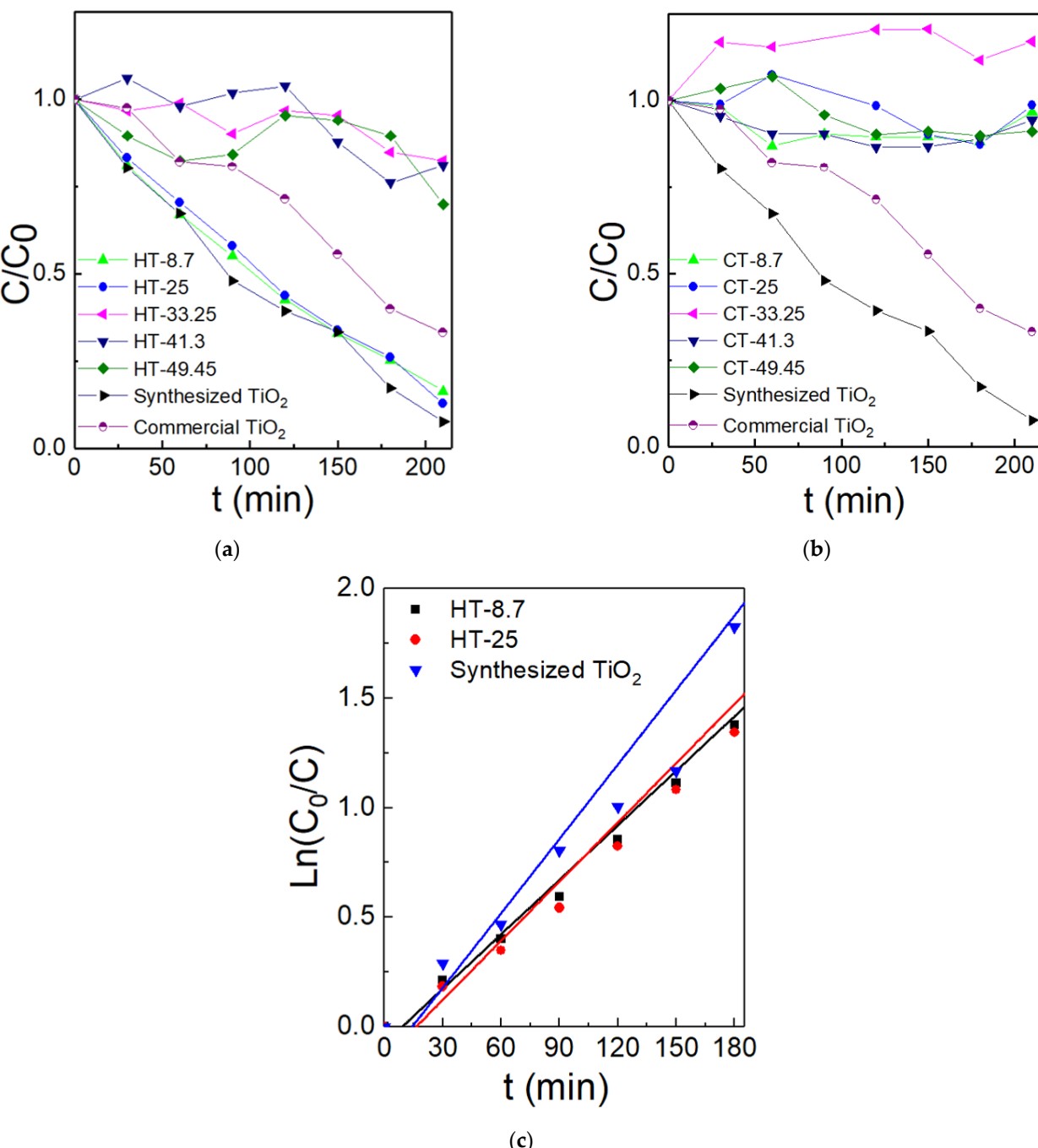

**Figure 14.** Time-dependent degradation of MO by photocatalytic activity of (**a**) HT and (**b**) CT composites. (**c**) The $\ln(C_0/C)$ vs. t plot.

According to the results obtained in this work and in the literature [43–49], the photocatalytic degradation of MO can be described as follows (Figure 15): when $TiO_2$ is loaded onto the substrate surface, it is illuminated by UV light, and the photogenerated electron-hole pair may then migrate to the surface. The electrons on the surface reduce the adsorbed oxygen molecules to $O_2^{\cdot-}$ radicals, and the photogenerated holes migrate toward the surface to oxidize the adsorbed water to form $\cdot OH$ species. The $O_2^{\cdot-}$ radicals, in turn, may generate $HO_2\cdot$ species. Due to the $Ti^{3+}$ states, this avoids the electron-hole pair recombination and increases the number of $O_2^{\cdot-}$ species generated in the system. The $O_2^{\cdot-}$, $\cdot OH$, and $HO_2\cdot$ radicals are very reactive and decompose the MO molecules adsorbed on the surface of the composite; this process is faster than the velocity of the desorption of the MO molecules,

allow for the degradation of the dye. The larger photocatalytic activity in HT-8.7 and HT-25 can be explained in terms of a larger crystallinity degree evidenced in the Raman results, a synergic effect between the substrate and the amount of $TiO_2$ that lies on a relatively high surface area, with a partial fill up of substrate pores evidenced by $N_2$ physisorption and DBAR measurements, in comparison with other composites.

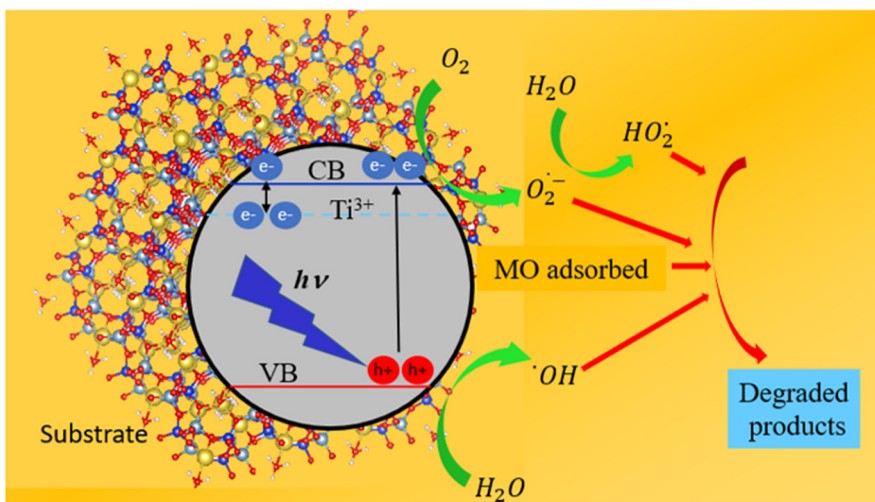

**Figure 15.** Scheme of photocatalytic mechanism for MO degradation.

## 3. Materials and Methods

### 3.1. Materials

CFA with a size smaller than 38 μm were selected by sieving, and were collected from the Sochagota TermoPaipa IV thermoelectric power plant (Boyacá, Colombia). NaOH (PanReac AppliChem 98%) and deionized water were employed to perform the alkaline activation of CFA. Titanium isopropoxide (Alfa Aesar 97+%) and ethanol (Alfa Aesar 96%) were used to synthetize $TiO_2$.

### 3.2. Properties of CFA

The chemical composition of CFA was determined by using a MagixPro PW-2440 Philips X-ray fluorescence spectrometer. The morphology of CFA was observed by using a Carl Zeiss EVO-MA10 electron microscope at the accelerating voltage of 10 kV. The crystalline phases were identified by using a PANalytical X'Pert's X-ray diffractometer with $\lambda_{K\alpha\text{-}Co} = 1.789007$ Å.

### 3.3. Alkaline Activation of CFA by the Hydrothermal Method

Two different samples, with larger conversions of CFA into zeolite and under conditions of a low synthesis temperature and concentration of NaOH, as described in previous works, were selected for use as substrates for $TiO_2$ immobilization [22,35]. Briefly, CFA was activated by the conventional hydrothermal method using a Teflon® vessel placed within a steel reactor and covered by a thermal insulation in order to keep the temperature at a constant value, which was monitored with a K thermocouple with the following synthesis parameters:

(a) 4.2 g of CFA was added in NaOH solution (3 M) at 99.5 °C for 24 h.
(b) 10 g of CFA was added in NaOH solution (4 M) at 95 °C for 24 h.

After the alkaline activation, the solution was filtered, washed with ethanol, filtered again, and dried. In addition, in this work, the as prepared samples were calcined at 500 °C. From a previous procedure, two samples were obtained with zeolite as the major crystalline phase, corresponding to hydroxysodalite, and cancrinite labeled as H and C, respectively; these samples were used as substrates to immobilize the $TiO_2$ particles.

*3.4. Synthesis of TiO$_2$-Zeolite Composites*

Wetness impregnation of TiO$_2$ over H and C samples was carried out as follows: H and C substrates were placed in ethanol under magnetic stirring at 200 rpm, titanium isopropoxide was added drop by drop with different nominal TiO$_2$ loading at 60 °C, and the solution was kept under magnetic stirring for 6 h. The as-prepared composites were dried at 90 °C overnight and calcined at 425 °C for 2 h. A sample of TiO$_2$ was synthesized following this procedure at 425 °C for 6 h. The obtained composites were labelled according to the scheme of Table 4, in which the HT nomenclature refers to TiO$_2$ (T) impregnated over the substrate with main phase of hydroxysodalite (H) to obtain the composite HT and, the CT label refers to TiO$_2$ (T) impregnated over the substrate with the main phase of cancrinite (C) to obtain the composite CT.

**Table 4.** Labeling of samples showing the amount of immobilized TiO$_2$.

| TiO$_2$ Loaded over H Substrate | TiO$_2$ Loaded over C Substrate | TiO$_2$ (wt%) |
|---|---|---|
| HT-8.7 | CT-8.7 | 8.70 |
| HT-25 | CT-25 | 25.00 |
| HT-33.25 | CT-33.25 | 33.25 |
| HT-41.3 | CT-41.3 | 41.30 |
| HT-49.45 | CT-49.45 | 49.45 |

*3.5. Characterization of TiO$_2$-Zeolite Composites*

The structural characterization was performed by X-ray diffraction (XRD) using a Bruker AXS D8 focus spectrometer with $\lambda_{Cu-K\alpha}$ = 0.15405 nm. The Raman spectra were obtained in a Horiba Yobin-Yvon T64000 dispersive Raman spectrometer using a laser source of 532 nm and a power of 0.001 W over the sample. Electron spin resonance (ESR) spectra were obtained at room temperature in a Bruker EMX 10-2.7 Plus spectrometer. The morphological properties were observed by high resolution transmission electron microscopy (HRTEM) in a Tecnai F20 Super Twin TMP microscope of FEI; low-temperature N$_2$ physisorption measurements were performed using Micromeritics ASAP 2020 equipment, with Doppler broadening annihilation radiation of the positrons, and energy measurements (DBAR) were performed in the slow positron beam facility at Washington State University, USA; the 511 keV annihilation peak was recorded using an ORTEC HPGe detector. The optical properties were studied by ultraviolet-visible diffuse reflectance spectroscopy (DRS) using an UV-VIS Cary 50 VARIAN spectrophotometer equipped with an integrating sphere; BaSO$_4$ was used as the reference material.

The photocatalytic behavior of the materials was studied in the degradation of MO dye (C$_{14}$H$_{14}$N$_3$NaO$_3$S). For this purpose, 200 mL of a MO aqueous solution at 25 ppm and 50 mg of photocatalytic material were added in a Pyrex reactor. The suspension was maintained under magnetic stirring at 700 rpm, and it was bubbled with constant air flow. The system was irradiated by employing a low-pressure mercury lamp (254 nm, 4.4 mW) protected with a quartz tube, which was immersed in the suspension. To favor the adsorption-desorption equilibrium, the reaction system was maintained in the dark for 30 min. Then, the irradiation was initiated, and aliquots, which were filtered by a membrane filter, were extracted every 30 min. The samples were analyzed by using an Evolution 300 UV-Vis spectrophotometer. Finally, the residual total organic carbon (TOC) was estimated using a TOC/TN analyzer multi N/C 2100.

## 4. Conclusions

Alkaline activation of CFA by the hydrothermal method under hydrothermal conditions allows for the obtaining of aluminosilicates with zeolite materials as majority phases, which were then used as substrates to immobilize TiO$_2$, whereby it was possible to produce TiO$_2$-zeolite composites at low cost. The synthesis method permits the obtaining of TiO$_2$ in an anatase phase, with a similar lattice parameter and a long-range of crystalline order

for most of the samples. The procedure performed to synthesize the TiO$_2$-composites allows for the obtaining of anatase TiO$_2$ with a Ti$^{3+}$ state, indicating the presence of oxygen vacancies without subsequent treatment. The textural properties of the composites were improved in comparison with those of the substrates, since, due to TiO$_2$ incorporation, the surface of the substrate was changed. The results verify that TiO$_2$ particles were located on the surface and within the pores of the substrate. On the surface, the particles of TiO$_2$ crystallize in a wide range of particle sizes due to the agglomeration effect. The HT-8.7, HT-25, and synthesized TiO$_2$ showed a degradation of MO in an aqueous medium under UV light, which was higher than that of commercial Degussa P25 TiO$_2$, due to a synergic effect between the substrate and the amount of TiO$_2$.

**Author Contributions:** Conceptualization, writing, formal analysis, investigation, I.S.G.; writing, investigation, formal analysis, resources, C.A.P.G.; writing, investigation, M.H.W.; investigation, I.M.S.G.; writing, investigation, formal analysis, C.P.C.M.; writing, formal analysis, J.J.M.Z.; resources, H.A.R.S.; writing, conceptualization, formal analysis, J.A.M.C.; resources, supervision, M.A.A.; writing, formal analysis, C.R.; conceptualization, resources, supervision, C.A.P.V.; conceptualization, resources, supervision, J.A.M.G. All authors have read and agreed to the published version of the manuscript.

**Funding:** This work was financed by the Gobernación de Boyacá (Grant No. 733 Colciencias), Universidad Antonio Nariño, under the project No PI/UAN-2021-687GFM and FAPESP (Grant No. 2017/10581-1).

**Institutional Review Board Statement:** Not applicable.

**Informed Consent Statement:** Not applicable.

**Data Availability Statement:** Not applicable.

**Acknowledgments:** We acknowledge CEM—Centrais Experimentais Multiusuário—UFABC for its support of this project.

**Conflicts of Interest:** The authors declare no conflict of interest.

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
