# Peer review of "Physicochemical Properties of Ti3+ Self-Doped TiO2 Loaded on Recycled Fly-Ash Based Zeolites for Degradation of Methyl Orange"

_condensedmatter, doi:10.3390/condmat7040069_

Round 1

Reviewer 1 Report

Below you will find my impressions concerning the article condensedmatter-1971531 “Physicochemical properties of Ti3+ self-doped TiO2 loaded on recycled fly-ash based zeolites for degradation of methyl orange” from Carlos Andrés Palacio Gómez at al.

Impression

In the reviewers' view, this manuscript is a well-written study with very well-conducted experiments that can be published after only minor revisions.

 Specific comments and suggestions

As already mentioned, I have only very minor corrections to note, which does not mean that I did not read the entire manuscript and enjoyed it.

 Line 39: I think you can replace “Pollution by organic dyes originated” by “Pollution by organic compounds originated”.

Line 44: “nontoxic” is hardly true for nanoscaled power material which is insoluble in nearly anything and can result in OH racicals.”less-harmful” is better from reviewers opinion.

 Line 144: I am not sure is it “XRD” or “RDX”?

 XRD: You used the hkl values in equation 2, so you might add them in the figures.

 ESR: The signal of the spin probe is very small compared to the measured signal, so the error of the calculated value is very large ? line 190

 Line 363ff

The degradation of MO miss the whole part of the Photo-Kolbe reaction (e.g. proven in studies of photo Kolbe ZnO) Photo semiconductors like TiO2 can in this way completely mineralize organic compounds. In the indicated references (in Photo-Curing of off-set Printing Inks by Functionalized ZnO from Hempelmann ) you will also find that the holes can oxidize C-OH to C=O, and COOH, CO2

 Line 418 (Calcination) Did you also analyses the samples by XRD before calcination?

 Line 447: Can you verify the lamp properties. Electrical imput power. Output mW/cm² (plus area). One line with 254 nm is a low-pressure lamp  (not high pressure). Do you have a mid-pressure or a high-pressure lamp. Would be best to verify the spectrum using a suitable Diode array detector.

Supporting information with pictures of the samples always are helpful. Did you observe a change in coloration of the samples due to the illumination (blue coloration of the powder)

Reviewer 2 Report

In this paper, TiO2-zeolite composites were synthesized using recycled coal flyash from a local thermoelectric power plant to produce the zeolite by hydrothermal method. The results show that TiO2 crystallizes in anatase phase with Ti3+ oxidation state without post-treatment. TiO2 particles were located within pores of the substrate and on its surface, increasing the surface area of composites in comparison with that of substrates. Samples with TiO2 at 8.7 and 25 wt% immobilized over hydroxysodalite show the highest degradation of methyl orange among all studied materials including the commercial TiO2 Degussa P25 under UV light. The result is interesting. The manuscript can be accepted after following revisions.

1 Ti3+ is not stable. Thus, please confirms the existence of Ti3+ in samples and indicate the role of Ti3+ during photocatalysis.

Reviewer 3 Report

Referee report on “Physicochemical properties of Ti3+ self-doped TiO2 loaded on recycled fly-ash based zeolites for degradation of methyl orange" by Iván Supelano García et al.

Although this topic is of some interest, this manuscript in its present form cannot be recommended for publication and requires at least some important improvement and justifications.

1.     Although the introduction is well written, several points are either not disclosed or not completely accurate. In particular, nothing is said about the morphology of TiO2-based materials and how morphology affects the functional properties. A lot of recent studies have been published in MDPI journals. It is known that TiO2 can be in different crystalline modifications, and the nanostructure is also very diverse, both in structure and size. Few of them are below:

Serga, V.; et al Crystals 202111, 431. https://doi.org/10.3390/cryst11040431

Lin, Y.-P.; et al. Nanomaterials 202111, 2900. https://doi.org/10.3390/nano11112900

Liu, Y. Materials 2022, 15, 5875. https://doi.org/10.3390/ma15175875

And references therein.

2.     Lines 96-99. This part requires analysis of measurement error. Please, specify error bars.

3.     Figure 1. How stable is this image over time? Is there aging?

4.     Last column of Table 1. Please, specify error bars.

5.     Fig. 5 (a, b). It would be useful to bring all the observed Raman peaks into one Table, where they should be given an additional interpretation with the corresponding literature data.

6.     It is not at all clear where the data in Table 2 came from. 2, e.g. Eg values and how they were obtained.

7.     Pore size accuracy (Table 2) is extremely unrealistic and needs to be justified.

 In general, the manuscript is interesting and can be recommended for publication after constructive reflection on the above comments.

Round 2

Reviewer 3 Report

In my previous report, the was the following comment:

Fig. 5 (a, b). It would be useful to bring all the observed Raman peaks into one Table, where they should be given an additional interpretation with the corresponding literature data.

 Below is the response of the anthor:  The observed Raman peaks are summarized between lines 162 and 164. For the purpose of the article, we don’t consider to give a dipper interpretation.

Let me copy of the sentence  between lines 162-164:

".... group of the anatase around 162 144, 197, 399, 519 and 638 cm-1 , corresponding to Eg(1), Eg(2), B1g(1), an overlap of B1g(2) 163 with A1g, and Eg(3) optical modes respectively [30,31] ".

It is important to note that reference [30] is about MgAl2O4 and in this case it is incorrect, while [31] is about Donor-Doped TiO2. 

I would like to ask once again to make a detailed comparison of these data with the Raman spectra of titanium oxide and, at the same time, take as a basis a single article for comparison.

Round 3

Reviewer 3 Report

Now after 2nd revision, this paper can be accepted